# Trends in Cancer Incidence and Mortality in US Adolescents and Young Adults, 2016–2021

**DOI:** 10.3390/cancers16183153

**Published:** 2024-09-14

**Authors:** Li Zhang, Joshua E. Muscat, Vernon M. Chinchilli, Chandrika G. Behura

**Affiliations:** 1Center for Research on Tobacco and Health, Department of Public Health Sciences, Penn State College of Medicine, Hershey, PA 17033, USA; lzhang9@pennstatehealth.psu.edu; 2Department of Public Health Sciences, Penn State College of Medicine, Hershey, PA 17033, USA; vchinchilli@pennstatehealth.psu.edu; 3Department of Pediatrics, Penn State Children’s Hospital, Penn State University College of Medicine, Hershey, PA 17033, USA

**Keywords:** cancer, AYAs, incidence, mortality, time trends, metropolitan, nonmetropolitan, race, SEER

## Abstract

**Simple Summary:**

The incidence and mortality rates of cancer in the Surveillance, Epidemiology, and End Results (SEER) Program for adolescent and young adult (AYA) patients show distinct patterns among early-onset cancers. For some cancers, AYA cancer rates varied by age group, sex, race, ethnicity and geography. Monitoring the rates and time trends of AYA cancer emphasizes the distinct health concern for this age group.

**Abstract:**

(1) Background: The incidence rate of early onset-cancer (<50) has increased since 1995. Among younger people, cancers in AYAs (aged 15–39 y) are often biologically distinct tumors from those treated in the pediatric and older adult population. The current study describes trends in the United States for the most recent years including the first year of the COVID-19 epidemic. We aimed to describe the recent incidence and mortality trends of cancers in AYAs (aged 15–39 y). (2) Methods: We used data from the Surveillance, Epidemiology, and End Results (SEER 22) from 1 January 2016 to 31 December 2021. Age-adjusted incidence and mortality rates were assessed by SEER*Stat 8.4.3 for major cancer types by sex, race/ethnicity, age, and metropolitan/nonmetropolitan status. Time trends of age-adjusted incidence and mortality rates were examined by sex and metropolitan/nonmetropolitan status. (3) Results: Age-adjusted overall cancer incidence and mortality rates were stable during this study period. The age-adjusted incidence rates declined significantly for ependymoma, melanoma, carcinomas of lung, bronchus, and trachea, unspecified malignant neoplasms, and non-Hodgkin’s lymphoma. Significant increases were found for gastrointestinal tract cancers and non-Kaposi sarcomas. The age-adjusted mortality rate decreased for acute myeloid leukemia, melanoma, carcinomas of liver and intrahepatic bile ducts, kidney and, in women, leukemia. For some cancers, rates differed by sex, race, ethnicity, and geography. Monitoring the rates and time trends of AYA cancer emphasizes the distinct health concern for this age group.

## 1. Introduction

Early-onset (<50) cancer incidence rates have increased in the United States and worldwide since the 1990s [1,2,3,4,5]. There are significant concerns that these trends may continue in the future, particularly for Generation X born between 1965–1980 [6,7]. Adolescents and young adults (AYAs) aged 15–39 with cancer are considered a unique population, and different from pediatric or older groups [8]. The recognition of AYA cancer as distinct early-onset cancer is based on the types of tumors and their biology that occur in this age group [8,9]. Additionally, the detection and management of AYA patients entail unique challenges ranging from delays in diagnosis, psychological support, fertility, and financial needs during their transition into older adulthood [8,10,11,12,13,14,15].

Cancer is the leading cause of mortality in AYAs, and comprises about 4.2% of all cancers in the NCI’s Surveillance, Epidemiology, and End Results (SEER) Program [16]. The incidence and types of cancer in AYAs are age-dependent, with Hodgkin’s lymphoma, and central nervous system tumors more common in adolescents, whereas thyroid, testicular and non-Hodgkin’s’ lymphoma are relatively common in the 20–29-year range [16,17]. Young adults aged 30–39 experience cancer types more similar to older young adults (40–49) including GI, melanoma, and breast cancer [16,17].

Their overall incidence and mortality trends from 1973–2015 are shown on SEER’s website [16]. Mortality rates slowly declined since 1992, while incidence rates steadily increased from about 1997 [16]. Monitoring changes in incidence and mortality rates is essential for understanding disease etiology and planning for future needs in the AYA population. During the first year of the COVID-19 epidemic in the USA, there was an increase in underdiagnosed cancer cases in the USA, and COVID-19 may have affected AYAs similarly [18]. The current study was conducted to characterize recent trends of AYA cancer (2016–2021) incidence and mortality rates, and examine these by cancer subtype, sex, race and ethnicity, age group, and metropolitan/nonmetropolitan designation.

## 2. Materials and Methods

### 2.1. Data Sources and Acquisition

SEER is a network of population-based cancer registries in geographic areas that represent the racial and ethnic populations in the US [19]. The SEER 22 database used for this analysis encompasses 22 USA regions, representing 47.9% of the US population [20]. The registries routinely collect data on incidence, mortality, primary tumor site, and tumor morphology, year of diagnosis, year of death, age at diagnosis, age at death, sex, race/ethnicity, metropolitan/nonmetropolitan status. SEER-imbedded cancer subtype coding specific for AYA site/histology codes (AYA site recode) was developed to better characterize topography and morphology in AYA and have undergone periodic revision [21]. SEER is the most comprehensive population-based source in the USA that provides de-identified information on cancer patients, which offers opportunity to study cancer epidemiology in AYAs in a convenient fashion without conducting high-cost epidemiological studies that involve blinding or randomization.

For the incidence or mortality analysis, the inclusion criteria included AYAs (age at diagnosis or at death: 15–39 y) with a diagnosis of invasive cancer or who had died from cancer from 1 January 2016 to 31 December 2021. Age, race/ethnicity, sex, and metropolitan/nonmetropolitan designation information were included for both analyses. We excluded all the cancer patients from SEER who were not AYAs (age at diagnosis or age at death <15 y or >39 y), who reported missing information on race/ethnicity, sex, metropolitan/nonmetropolitan designation, age, or the ones whose cancer diagnosis or death occurred before 1 January 2016, or after 31 December 2021.

Both analyses included sociodemographic information of the AYA patients such as age, sex, race/ethnicity, and metropolitan/nonmetropolitan status. Weight information was not included because SEER does not collect weight information of cancer patients. Age ranged from 15–39 years and was categorized into five groups: 15–19, 20–24, 25–29, 30–34, and 35–39. Sex was grouped into female and male. Race/ethnicity group was classified as non-Hispanic White (NHW), non-Hispanic black (NHB), non-Hispanic American Indian/Asian Pacific (NHAIAP), and Hispanic. Metropolitan/nonmetropolitan status was designated based on USDA’s 2013 rural-urban continuum codes (RUCC) at county level imbedded in SEER*Stat [22,23]. Metropolitan counties were distinguished by population size of the metro area (RUCC code 1-3), while nonmetropolitan counties (RUCC code 4-9) by size of the largest city or town and proximity to metro and micropolitan areas (U.S. metropolitan/nonmetropolitan population: 85.9% vs. 14.1%) [24]. In both analyses, the AYAs with missing/unknown metropolitan/nonmetropolitan status from Alaska and/or Hawaii were included as a group of the metropolitan/nonmetropolitan status because this information has been reported in SEER*Stat. However, this group was excluded from the trend analysis stratified by metropolitan/nonmetropolitan designation for its small sample size, which may not generate sufficient statistical power to show significant trend change.

SEER*Stat 8.4.3 was used to calculate age-adjusted incidence and mortality rates and measure annual percentage changes (APC) in age-adjusted cancer incidence and mortality rates as well as significant trends in these rates [25]. Comparison of the rates by race, sex, and metropolitan/nonmetropolitan designation were also calculated using SEER*Stat 8.4.3. Similarly, the age-adjusted incidence rates comparison between two periods: 2020–2021 and 2016–2019 (reference) was done in cancer subtypes. The significance was estimated using the exact mid-*p* double sided *p*-value. The current study was exempt from the institutional review board oversight of Penn State for using de-identified and publicly available data from SEER.

### 2.2. Statistical Analysis

Two SEER methods are used for the calculation of cancer rates and time trends. When an individual case has multiple diagnoses, one method is to calculate the rate based on only the first record for these individuals. The second method counts every cancer diagnosis. We determined that these two methods have a 98% agreement rate. All age-adjusted AYA cancer incidence and mortality rates, reported per 100,000 persons, were calculated using SEER*Stat, version 8.4.3 and age-adjusted to the 2000 US standard population (19 age groups, P25-1130) [26]. Mortality cases were retrieved from cancer registry-determined vital statistics collected as follow-up from the SEER’s incidence-based mortality database [27,28]. Percent changes from 2016 to 2021 in age-adjusted incidence rates and in mortality rates were analyzed by fitting linear regression models with each calendar year at diagnosis or year of death (2016–2021) as an independent variable. Percent changes (significant increasing/decreasing or stable) in age-adjusted incidence and mortality rates from 2016–2021 were assessed using APC calculated based on weighted least square methods and corresponding confidence intervals (CIs) estimated by SEER*Stat 8.4.3. The APC values allow for comparisons between incidence and mortality rate changes in rare and common cancers. Statistically significant APCs were different from zero with two-sided *p* < 0.05 [25]. Age-adjusted rates, time trends, and rate ratios were determined by cancer subtypes, sex, race/ethnicity, AYA age group, and metropolitan/nonmetropolitan status. For the completeness of data update, crude counts were also provided using SEER*Stat 8.4.3. Because SEER provides all these de-identified data on its website upon request, our findings can be replicated if all the same inclusion and exclusion criteria are adopted. The mortality-to-incidence ratio (MIR), a proxy for AYA cancer survival, was calculated by dividing the age-adjusted mortality rate by the age-adjusted incidence rate, for the selected cancers.

## 3. Results

### 3.1. Study Population Characteristics

This study included 229,975 AYAs (aged 15–39 y) diagnosed with invasive cancer (women: 141,091 and men: 88,884) and 29,400 AYAs died from invasive cancer (men: 14,533 and women: 14,867) from 2016 to 2021 (Table 1 and Appendix A). The majority of AYAs with cancer diagnosis were NHW (54%), followed by Hispanic (26.1%), NHB (10.6%), and NHAIAP (9.3%) (Table 2). There was a similar racial and ethnic distribution in cancer mortality (Table 2 and Appendix A). Most cases occurred in metropolitan areas (metropolitan: 91.3% and nonmetropolitan: 8.6%).

### 3.2. Age-Adjusted AYA Cancer Incidence Rates and Mortality Rates and MIR

From 2016–2021, the overall age-adjusted incidence rate for AYA was 72.3, and the overall mortality rate was 10.7 (Table 1). Thyroid and gastrointestinal tract carcinomas were the most common incident cases (15.3%) and cancer death (19.2%). Age-adjusted cancer incidence and mortality rates differ in patterns by cancer subtypes. The most common age-adjusted incidence rates were for breast cancer (11.0), thyroid carcinoma (10.9), lymphoma (7.6), carcinoma of gastrointestinal tract (7.5), male testis cancer (5.6), melanoma (5.2), carcinoma of genital sites (5.1), and leukemia (4.6). By contrast, the higher mortality rates were for carcinoma of gastrointestinal tract (2.1), breast cancer (1.2), leukemia (1.1), lymphomas (0.9), astrocytoma (0.9), sarcomas (0.9), and testis cancer (0.4). Melanoma (0.058) and gonadal and related tumors (0.096) had the relatively lower MIR.

### 3.3. Sociodemographic Differences in AYA Cancer Incidence and Mortality Rates

The age-adjusted incidence rate was significantly higher in females than males (90.9 vs. 54.3, *p* = 0.027) (Table 2). For males, the common new cases were carcinomas (32.5%). Of these, the most common were testis (21.0%), lymphoma (15.3%), leukemia (9.3%), and melanoma (6.5%) (Appendix A). The age-adjusted incidence rate (IR) for testicular cancer was 11.0. For females, carcinoma of breast (23.8%, IR: 22.3), thyroid carcinoma (20.2%), lymphoma (7.9%), melanoma (7.5%), and leukemia (4.7%) were the most common cancers.

The age-adjusted mortality rate was significantly higher in females than males (11.1 vs. 10.3) (Table 2). Stratified by sex, the main causes of death were carcinomas (34.4% and 59.5%), leukemias (13.6% and 8.6%), CNS and other intracranial and intraspinal neoplasms (12% and 7.6%), lymphomas (11% and 5.9%), and sarcomas (10.7% and 7.5%, Appendix A). Carcinomas of the breast (2.5) and gastrointestinal tract (2.3) topped mortality rates for female and male AYAs, respectively.

Table 2 also shows the rates by race and ethnicity. The highest age-adjusted incidence rate was among NHW (82.3), and the lowest was among NHAIAP (59.9). While NHAIAP had the lowest overall mortality rate (8.5), NHB had the highest rate (14.4). For females, the age-adjusted incidence rate in NHW was 102.3, which was significantly higher than other races. Similar differences were found for males. NHB had significantly higher mortality rates for both sexes.

Age-adjusted incidence and mortality rates significantly differed by geography (Table 2). The overall age-adjusted incidence and mortality rates were both significantly lower for metropolitan areas (incidence rate: 72.0, mortality rate: 10.5) as compared to nonmetropolitan areas (incidence rate: 76.0 and mortality rate: 12.4). When stratified by sex, the geographic difference in age-adjusted incidence rates was found only in women (Appendix A).

Categorized by age group, the findings of the sociodemographic distribution of cancer incidence and mortality rates were mostly consistent, showing higher incidence rates in females vs. males, NHW vs. other racial groups (higher mortality was with NHB), nonmetropolitan vs. metropolitan, with differing patterns in certain age groups (Table 2).

### 3.4. AYA Age-Adjusted Cancer Incidence Rates and Mortality Trends

Table 3 displays the age-adjusted incidence and mortality trends from 2016–2021 for the major histologic classifications and primary site. There was no significant percent change in the overall age-adjusted cancer incidence rates (APC = −1.2, 95% CI: −3.1, 0.8, *p* = 0.18) and mortality (APC = −0.5, 95% CI: −1.0, 0.1, *p* = 0.07) during the study period. The age-adjusted incidence rate increased for non-Kaposi sarcoma (APC = 8.3, *p* = 0.02), carcinoma of rectum (APC = 2.7, *p* = 0.01), and gastrointestinal tract (APC = 1.1, *p* = 0.04). There was a decrease in the incidence of ependymoma (−5.4, *p* = 0.047), melanoma (−5.1, *p* = 0.02), carcinoma of lung, bronchus, and trachea (LBT) (−4.5, *p* = 0.04), unspecified malignant neoplasms (UMN) (−4.3, *p* = 0.01), liposarcoma (−4.2, *p* = 0.01), and NHL (−2.6, *p* = 0.002). A decrease in mortality was found for AML (APC = −3.4, *p* = 0.047), melanoma (−7.4, *p* = 0.047), carcinomas of liver and intrahepatic bile duct (IBD) (−4.1, *p* = 0.02), and kidney (−6, *p* = 0.02). The death rates increased for UMN (3.5, *p* = 0.046).

### 3.5. Sociodemographic Differences in AYA Cancer Incidence and Mortality Trends

When stratified by sex, there were no significant changes in age-adjusted incidence rates for both males (APC = −1.3, *p* = 0.11) and females (APC = −1.1, *p* = 0.25) in this period (Appendix A). When stratified by type, there was a decreasing trend for melanoma in both men and women (males: −6.4, *p* = 0.01; females: −4.3, *p* = 0.032). For males only, age-adjusted cancer incidence rates decreased for NHL (−2.8, *p* = 0.004) and fibromatous neoplasms (−6.8, *p* = 0.04), but increased in non-Kaposi sarcoma (9.3, *p* = 0.04). For females alone, age-adjusted incidence rates increased for ALL (APC = 3.4, *p* = 0.01) and blood and lymphatic vessel tumors (7.7, *p* = 0.003), but decreased for ependymoma (−5.9 *p* = 0.03), lymphoma (−1.6, *p* = 0.03), GCT (−5.8, *p* = 0.01), UMN (−6.5, *p* = 0.009), and synovial sarcoma (−4.4, *p* = 0.04).

The overall cancer mortality rates significantly decreased for females (APC = −0.8, *p* = 0.02). In women, a significant reduction was found for leukemia (−2.6, *p* = 0.03). In men, there was a decline in mortality from leukemias (−1.4, *p* = 0.01), NHL (−5.9, *p* = 0.03), and kidney carcinoma (−4.5, *p* = 0.03). Male mortality increased significantly for other and unspecific leukemia (11.9, *p* = 0.01) and nerve sheath tumors (15.4, *p* = 0.047). The rates and trends were more pronounced when calculated using the method that accounts for multiple cancer diagnosis and cancer deaths (Appendix A).

A significant decline of incidence rates in 2020–2021, as compared to 2016–2019, was found in AYAs for NHL (3.38 vs. 3.65), astrocytoma (2.31 vs. 2.43), glioblastoma (0.47 vs. 0.53), melanoma (4.39 vs. 5.3), carcinoma of the thyroid (10.0 vs. 11.07), head and neck (1.34 vs. 1.45), oral cavity, lip, and pharynx (0.65 vs. 0.72), LBT (0.69 vs. 0.83), and genital sites (4.62 vs. 4.98), as well as UMN (0.51 vs. 0.59).

Appendix A depicts the metropolitan/nonmetropolitan differences in cancer incidence trends. The overall age-adjusted incidence rates were stable for both metropolitan and nonmetropolitan areas (metropolitan: APC = −1.15, *p* = 0.2; nonmetropolitan: APC = −1.48, *p* = 0.07). Both metropolitan and nonmetropolitan areas had an increase in non-Kaposi sarcoma (metropolitan: APC = 6.4, *p* = 0.02; nonmetropolitan: APC = 34.94, *p* = 0.009). Incidence of rectum carcinoma also increased in metropolitan areas, while ovary carcinoma incidence increased significantly in nonmetropolitan AYAs. Further, NHL (APC = −2.6, *p* = 0.004), ependymoma (APC = −5.2, *p* = 0.045), liposarcoma (APC = −4.4, *p* = 0.01), melanoma (APC = −5.5, *p* = 0.01), carcinomas of LBT (APC = −4.9, *p* = 0.04), uterine cervix (APC = −3.3, *p* = 0.04), and UMN (APC = −4.6, *p* = 0.01) significantly declined in metropolitan areas. In nonmetropolitan areas, a significant decrease was found for fibromatous neoplasm (APC = −8.5, *p* = 0.008), thyroid carcinoma (APC = −6.7, *p* = 0.045), carcinomas of skin (APC = −14.1, *p* = 0.02), urinary tract (APC = −5, *p* = 0.03), and kidney (APC = −5.6, *p* = 0.02).

The mortality rates were stable in both metropolitan and nonmetropolitan areas (metropolitan: APC = −0.35, *p* = 0.297; nonmetropolitan: APC = −1.78, *p* = 0.342, Appendix A). In metropolitan areas, mortality rates declined for AML (APC = −3.5, *p* = 0.01) and carcinoma of kidney (APC = −5.6, *p* = 0.03), but increased for male testis (APC = 6.2, *p* = 0.02), carcinoma of esophagus (APC = 9.2, *p* = 0.02), and gonadal and related tumors (APC = 3.2, *p* = 0.04). For nonmetropolitan areas, the mortality rate fell for ependymoma (APC = −21.8, *p* = 0.4), but rose for glioblastoma (APC = 12.1, *p* = 0.04) and pancreas carcinoma (APC = 14.6, *p* = 0.047). The mortality of nerve sheath tumors differed between metropolitan and nonmetropolitan areas, with an increase in metropolitan areas (APC = 16.7, *p* = 0.03), and a fall in nonmetropolitan areas (APC = −23.2, *p* = 0.009). The results were similar when considering multiple primary cancer diagnoses and deaths (Appendix A).

## 4. Discussion

Age-adjusted AYA cancer incidence rates significantly increased by 29.6% from 57.2 to 74.2 between 1973 and 2015 [17]. After rising for over 40 years, the age-adjusted incidence rates of overall cancer were stable as reported in our study between 2016–2021. This may reflect the stable incidence trends of carcinomas of the testis, thyroid, and kidney that were driving the previous rising incidence rates [7,17,29]. The falling incidence trend of melanoma and male NHL reported in our study contradicted the previous finding where the melanoma incidence rates increased drastically, and the trend of male NHL was stable from 1973–2015 [17]. Additionally, we reported a decrease in age-adjusted incidence rates of carcinoma of LBT, while a decline in carcinoma of LBT was only evident in male or female AYAs from the majority of previous data [30,31]. Sex differences in LBT incidence remain to be elucidated [32].

Environmental exposure to poly- and perfluoroalkyl substances, endocrine disruptors, genetic susceptibility, low-birth weight, and suboptimal maternal age may play a role in testis cancer etiology [33,34,35,36]. Thyroid and kidney carcinomas are obesity related [37,38]. Recommendations against screening asymptomatic patients in 2017 could help curb the elevating thyroid incidence rates, stabilizing the incidence rates of thyroid carcinoma in our study [39].

In addition, it may be that changes in modifiable lifestyle risk factors in the population impact these trends. Melanomas tend to affect individuals with light skin and excessive UV exposure [40,41]. The increased use of preventive measures (staying in the shade and wearing long-sleeved clothes) from 2003–2018 in AYAs (59% were NHW) may help shield them from UV exposure, contributing to a decline in melanoma incidence rates [42]. Stratified by sex, a sharp decline in HIV infection rate in young adults (up to 34% reduction during 2017–2021) due to improved prevention, testing, and treatment of HIV might have contributed to a decline in NHL incidence in men [43]. The reported decline in male MHL was also consistent with the findings from the comparison of a incidence trend analysis between Generation X and prior generations [7].

The main effect of COVID-19 in the general population is a decrease in major cancers that can be detected by screening including lung, colon, female breast, and melanoma [44]. This indicates that cancer incidence may be underreported due to pandemics. There was also a decline of melanoma and lung cancer incidence in our study. The decline of other cancer incidence reported in our study likely reflects the impact that COVID-19 had on access to health care and cancer detection.

Age-adjusted incidence rates of all cancers are higher in metropolitan populations, although rates are declining faster in metropolitan than nonmetropolitan areas in the U.S. [45]. Similarly, this study found significantly higher age-adjusted incidence rates of AYA in metropolitan populations. However, no metropolitan/nonmetropolitan difference was reported in AYA cancer incidence time trends. Poor access to health services with imaging utilization and long distances to care facilities could negatively affect nonmetropolitan AYAs’ screening rates, contributing to the decreasing trend of thyroid carcinoma incidence in nonmetropolitan areas [46,47,48].

The mortality rate of 10.7 per 100,000 (with leading causes of death from leukemia, sarcomas, testis tumor, and breast tumor) in the current study for 2016–2021 was higher than for 2012–2016 [49]. The higher rate may be due to an increase in UMN mortality rates since all other types were either stable or falling. This study reported no significant change in the overall mortality rate, which could be attributable to the relatively small contributions (0.2–4% of total mortality counts) of cancer types that showed significant change. Decline in melanoma mortality rate was consistent with the national data (1986–2016), indicating that novel systematic therapies contribute to the decrease [50]. The stable leukemia incidence trend and the declined mortality likely suggest recent treatment advances such as improved treatment regimens from tyrosine kinase inhibitors and allogeneic hematopoietic stem cell transplant [51,52].

This study also found sex differences in overall and site-specific mortality trends. Overall cancer mortality rates significantly decreased 5% in females (largely driven by a decrease in leukemia mortality) whereas the overall mortality rates were stable in male AYAs. While males had significant increase in mortality rates in other and unspecified leukemia and nerve sheath tumors, and decrease in leukemias, NHL, and carcinoma of kidney, the contribution of each of these cancer sites was small (<14%).

Our study has limitations. RUCC (2013) may not fully represent the contemporary picture of geographical regions during 2020–2021 for not capturing the changes made by the U.S. Census Bureau in 2020 to reflect urban area population [23]. However, using SEER provides an excellent opportunity to examine cancer morbidity and mortality in AYAs as a distinctive population.

## 5. Conclusions

This study provides a recent profile of AYA cancer rates and trends. Unlike the increasing rates in adult-onset cancer, the incidence in AYAs has remained unchanged in recent years. Specific cancers need continued monitoring, and there is a need to identify factors that contribute to the disparities by sex, race/ethnic distribution, and geography.

## Figures and Tables

**Table 1 cancers-16-03153-t001:** Age-adjusted AYA cancer incidence rates (age at diagnosis, 15–39 y), and mortality rates for death from cancer (age at death, 15–39 y) from SEER-22, 2016–2021.

Cancer Subtype	Incidence			Mortality			MIR
	Rate ^1^	Count	%	Rate ^1^	Count	%	
1. Hematological malignancies	4.6	14,861	6.4	1.1	3253	11.1	0.239
1.1 Acute lymphoid leukemia	1.1	3663		0.5	1300		
1.2 Acute myeloid leukemia	1.2	3841		0.4	1181		
2. Lymphomas	7.6	24,821	10.8	0.9	2473	8.4	0.118
2.1 non-Hodgkin lymphoma	3.6	11,684		0.5	1513		
2.2 Hodgkin lymphoma	3.3	11,131		0.2	667		
3. Brain tumors	2.7	8677	3.8	1.0	2887	9.8	0.37
3.1 Astrocytoma (Astroglia and related neoplasms)	2.4	7856		0.9	2524		
4. Sarcomas	3.1	10,067	4.4	0.9	2670	9.1	0.375
4.1 Osteosarcoma	0.4	1187		0.2	561		
4.2 Chondrosarcoma	0.2	609		0.0	71		
4.3 Bone tumors	0.3	839		0.1	330		
4.4 Soft tissue tumors	0.4	1447		0.2	513		
4.5 Fibromatous neoplasms	0.5	1758		0.0	74		
4.6 Gastrointestinal stromal tumor	0.2	575		0.0	38		
5. Blood and lymphatic vessel tumors	0.5	1738	0.8	0.2	561	1.9	0.4
5.1 Specified (non-Kaposi sarcoma)	0.1	446		0.1	142		
5.2 Kaposi sarcoma	0.4	1292		0.1	419		
6. Nerve sheath tumors(malignant)	0.1	475	0.2	0.1	213	0.7	1.0
7. Gonadal and related tumors	7.3	24,235	10.5	0.7	1943	6.6	0.096
7.1 Testis	5.6	18,668		0.4	1028		
7.2 Ovary	1.4	4322		0.2	622		
7.3 Germ cell and trophoblastic (GCT)-CNS	0.1	363		0.0	70		
7.4 GCT (excluding CNS, testis, ovary)	0.3	874		0.1	221		
8. Melanoma	5.2	16,415	7.1	0.3	749	2.5	0.058
9. Carcinomas	40.3	125,960	54.8	5.1	13,847	47.1	0.127
9.1 Thyroid carcinoma	10.9	35,174		0.1	356		
9.2 Carcinoma of head and neck	1.5	4625		0.2	653		
9.2.1 Nasopharyngeal carcinoma	0.2	603		0.0	108		
9.2.2 Lip, oral cavity and pharynx carcinoma	0.7	2284		0.1	359		
9.2.3 Salivary gland	0.4	1179		0.0	76		
9.3 Carcinoma of gastrointestinal tract	7.5	23,381		2.1	5636		
9.3.1 Carcinoma of esophagus	0.1	445		0.1	212		
9.3.2 Carcinoma of stomach	0.8	2386		0.4	1129		
9.3.3 Carcinoma of small intestine	0.3	794		0.0	111		
9.3.4 Carcinoma of colon	3.3	10,348		0.6	1665		
9.3.5 Carcinoma of rectum	1.6	4961		0.4	965		
9.3.6 Carcinoma of anus	0.2	552		0.0	107		
9.3.7 Carcinoma of liver and IBD	0.4	1396		0.2	669		
9.3.8 Carcinoma of gallbladder and other extrahepatic biliary	0.1	406		0.1	160		
9.3.9 Carcinoma of Pancreas	0.6	1943		0.2	532		
9.4 Carcinoma of lung, bronchus, and trachea	0.8	2578		0.3	828		
9.5 Carcinoma of skin	0.1	262		0.0	18		
9.6 Carcinoma of breast	11.0	33,719		1.2	3234		
9.7 Carcinoma of genital sites (non ovary or testis)	5.1	15,946		0.7	1853		
9.7.1 Carcinoma of uterine cervix	3.0	9373		0.5	1422		
9.8 Carcinoma of urinary tract	2.9	8830		0.3	710		
9.8.1 Carcinoma of kidney	2.3	7050		0.2	543		
9.8.2 Carcinoma of bladder	0.5	1691		0.1	144		
9.9 Carcinoma of other and ill-defined sites, NOS	0.5	1445		0.2	559		
10. Unspecified Malignant Neoplasms, except CNS	0.6	1849	0.8	0.2	550	1.9	0.333
Any cancer type, Total	72.3	229,975	100	10.7	29,400	100	

^1^ All incidence and mortality rates were age-adjusted and reported per 100,000 persons.

**Table 2 cancers-16-03153-t002:** AYA cancer demographics for 2016–2021 by age group.

	Incidence								
	All			15–19 y			20–24 y		
Characteristics	Rate	Count	%	Rate	Count	%	Rate	Count	%
Sex									
Male	54.3	88,884	38.6	23.9	7802	52.3	35.5	11,846	50.2
Female	90.9 *	141,091	61.4	22.7 *	7105	47.7	37.1 *	11,738	49.8
Race/ethnicity									
Non-Hispanic White	82.3	124,164	54.0	25.5	7383	49.5	41.8	12,491	53.0
Non-Hispanic Black	61.0 *	24,370	10.6	16.8 *	1422	9.5	23.5 *	2090	8.9
Non-Hispanic American Indian/Asian Pacific	59.9 *	21,332	9.3	20.7 *	1179	7.9	31.5 *	1934	8.2
Hispanic (All Races)	65.8 *	60,109	26.1	23.6 *	4923	33.1	35.1 *	7069	29.9
Metropolitan status									
Metropolitan	72.0	209,909	91.3	23.4	13,557	90.9	36.4	21,537	91.3
Nonmetropolitan	76.0 *	19,832	8.6	22.3	1335	9.0	34.6 *	2023	8.6
Unknown/missing (Alaska or Hawaii)	87.1 *	234	0.1	23.0	15	0.1	39.0	24	0.1
Total	72.3	229,975	100	23.3	14,907	100	36.3	23,584	100
	25–29 y			30–34 y			35–39 y		
	Rate	Count	%	Rate	Count	%	Rate	Count	%
Sex									
Male	48.9	16,964	42.6	66.3	22,510	36.2	91.0	29,762	33.3
Female	68.7 *	22,863	57.4	121.0 *	39,718	63.8	187.5 *	59,667	66.7
Race/ethnicity									
Non-Hispanic White	67.8	21,463	53.9	106.9	34,172	54.9	156.2	48,655	54.4
Non-Hispanic Black	43.1 *	4036	10.1	77.6 *	6555	10.5	131.2 *	10,267	11.5
Non-Hispanic American Indian/Asian Pacific	46.2 *	3466	8.7	74.7 *	5916	9.5	116.4 *	8837	9.9
Hispanic (All Races)	55.8 *	10,862	27.3	84.4 *	15,585	25.1	120.6 *	21,670	24.2
Metropolitan status									
Metropolitan	58.4	36,512	91.1	92.6	56,940	91.5	137.5	81,363	91.0
Nonmetropolitan	60.3	3271	8.7	99.5 *	5216	8.4	150.1 *	7987	8.9
Unknown/missing (Alaska or Hawaii)	70.3	44	0.2	125.8 *	72	0.1	163.3 *	79	0.1
Total	58.6	39,827	100	93.2	62,228	100	138.6	89,429	100
	Mortality								
	All			15–19 y			20–24 y		
	Rate	Count	%	Rate	Count	%	Rate	Count	%
Sex									
Male	10.3	14,533	49.4	4.0	1156	60.9	5.6	1629	61.7
Female	11.1 *	14,867	50.6	2.7 *	742	39.1	3.7 *	1010	38.3
Race/ethnicity									
Non-Hispanic White	10.0	12,628	43.0	3.0	716	37.7	3.8	957	36.3
Non-Hispanic Black	14.4 *	4982	16.9	3.8 *	278	14.7	5.0 *	387	14.7
Non-Hispanic American Indian/Asian pacific	8.5 *	2729	9.3	3.2	167	8.8	3.7	207	7.8
Hispanic (All Races)	10.8 *	9061	30.8	3.8 *	737	38.8	5.9 *	1088	41.2
Metropolitan status									
Metropolitan	10.5	26,498	90.1	3.4	1726	90.9	4.7	2425	91.9
Nonmetropolitan	12.4 *	2845	9.7	3.1	167	8.8	4.0 *	211	8.0
Unknown/missing (Alaska or Hawaii)	21.3 *	57	0.2	7.7	5	0.3	4.9	3	0.1
Total	10.7	29,400	100	3.4	1898	100	4.6	2639	100
	25–29 y			30–34 y			35–39 y		
	Rate	Count	%	Rate	Count	%	Rate	Count	%
Sex									
Male	8.1	2461	56.1	12.0	3569	47.2	19.9	5718	44.3
Female	6.6 *	1928	43.9	13.9 *	3998	52.8	25.8 *	7189	55.7
Race/ethnicity									
Non-Hispanic White	6.6	1772	40.3	12.5	3364	44.5	22.1	5819	45.1
Non-Hispanic Black	8.6 *	702	16.0	17.6 *	1307	17.3	33.5 *	2308	17.9
Non-Hispanic American Indian/Asian pacific	6.2	422	9.7	9.7 *	694	9.1	18.0 *	1239	9.6
Hispanic (All Races)	8.3 *	1493	34.0	12.9	2202	29.1	21.4	3541	27.4
Metropolitan status									
Metropolitan	7.2	3965	90.3	12.7	6825	90.2	22.3	11,557	89.5
Nonmetropolitan	8.4 *	410	9.3	15.5 *	728	9.6	27.9 *	1329	10.3
Unknown/missing (Alaska or Hawaii)	22.4 *	14	0.4	24.5 *	14	0.2	43.4 *	21	0.3
Total	7.4	4389	100	12.9	7567	100	22.8	12,907	100

* Significant at 0.05 level. All incidence and mortality rates were age-adjusted and reported per 100,000 persons. SEER*Stat reports AYAs with missing/unknown metropolitan/nonmetropolitan status from Alaska or Hawaii.

**Table 3 cancers-16-03153-t003:** Trends in age-adjusted cancer incidence and mortality rates, 2016–2021.

Cancer Subtype	Incidence		Mortality	
	APC ^1^ (95% CI)	*p* Value	APC ^1^ (95% CI)	*p* Value
1. Leukemias	−0.1 (−2.8, 2.6)	0.90	−1.9 (−2.8, −1.0)	0.005
1.1 Acute lymphoid leukemia	1.9 (−0.5, 4.3)	0.09	−1.5 (−4.5, 1.6)	0.24
1.2 Acute myeloid leukemia	−0.1 (−5.2, 5.2)	0.95	−3.4 (−6.6, −0.1)	0.047
1.3 Chronic myeloid leukemia	−1.5 (−3.8, 0.8)	0.14	−2.8 (−10.9, 6.2)	0.43
1.4 Other and unspecified leukemia	0.8 (−5.3, 7.2)	0.75	7.5 (−4.2, 20.6)	0.16
2. Lymphomas	−1.6 (−3.0, −0.3)	0.03	−1.6 (−5.2, 2.1)	0.30
2.1 Non-Hodgkin lymphoma	−2.6 (−3.7, −1.5)	0.002	−3.7 (−8.4, 1.2)	0.10
2.2 Hodgkin lymphoma	−0.3 (−2.7, 2.1)	0.73	5.0 (−1.6, 12.0)	0.11
3. CNS and Other Intracranial and Intraspinal Neoplasms (All behavior)	−1.4 (−2.8, 0.1)	0.06	−0.3 (−4.7, 4.4)	0.88
3.1 Astrocytoma (Astroglia and related neoplasms)	−1.2 (−3.0, 0.7)	0.16	−0.1 (−4.4, 4.6)	0.97
3.1.1 Oligodendrogliomas	0.4 (−4.2, 5.2)	0.83	0.4 (−3.6, 4.6)	0.80
3.1.2 Glioblastoma (invasive)	−3.0 (−10.7, 5.4)	0.37	−0.5 (−4.9, 4.1)	0.77
3.1.3 Ependymoma (invasive)	−5.4 (−10.5, −0.02)	0.047	1.7 (−10.2, 15.2)	0.73
3.1.4 Other astrocytoma/astroglial neoplasms	−0.2 (−3.2, 2.8)	0.84	−0.02 (−6.2, 6.5)	0.99
3.2 Medulloblastoma	−2.3 (−7.0, 2.7)	0.27	2.2 (−6.3, 11.4)	0.52
4. Sarcomas	−2.5 (−4.8, −0.3)	0.04	−0.1 (−2.1, 2.0)	0.93
4.1 Osteosarcoma	−1.8 (−5.7, 2.1)	0.27	−1.3 (−8.2, 6.1)	0.64
4.2 Chondrosarcoma	−1.2 (−8.9, 7.1)	0.69	2.5 (−15.7, 24.5)	0.75
4.3 Ewing tumor	−2.0 (−6.1, 2.3)	0.26	−0.6 (−9.0, 8.5)	0.85
4.3.1 bone tumors	−1.0 (−3.7, 1.7)	0.36	−3.0 (−9.9, 4.5)	0.32
4.3.2 Soft tissue sarcomas	−3.5 (−13.8, 8.1)	0.44	4.2 (−9.9, 20.4)	0.48
4.4 Fibromatous neoplasms	−8.1 (−16.8, 1.5)	0.08	−7.6 (−19.6, 6.1)	0.19
4.5 Liposarcoma	−4.2 (−6.7, −1.6)	0.01	−0.3 (−19.3, 23.2)	0.97
4.6 Synovial sarcoma	−3.1 (−7.5, 1.6)	0.14	0.8 (−8.3, 10.9)	0.82
4.7 Leiomyosarcoma	−5.5 (−12.6, 2.1)	0.11	−9.1 (−18.9, 1.9)	0.08
4.8 Rhabdomyosarcoma	1.3 (−4.8, 7.8)	0.59	4.7 (−4.4, 14.7)	0.23
4.9 Gastrointestinal stromal tumor	9.9 (−6.6, 29.2)	0.18	1.9 (−21.0, 31.5)	0.84
4.10 Other soft tissue sarcoma	−1.7 (−6.3, 3.2)	0.39	3.5 (−3.5, 10.9)	0.25
4.11 Other bone tumors	−1.9 (−9.4, 6.3)	0.55	11.4 (−0.7, 24.9)	0.06
5. Blood and lymphatic vessel tumors	0.4 (−6.4, 7.7)	0.87	−1.5 (−5.3, 2.6)	0.36
5.1.1 Specified (non-Kaposi sarcoma)	8.3 (2.6, 14.3)	0.02	10.9 (−7.4, 32.9)	0.19
5.1.2 Kaposi sarcoma	−2.1 (−9.6, 6.0)	0.50	−4.9 (−11.5, 2.2)	0.12
6. Nerve sheath tumors(malignant)	1.9 (−3.0, 7.0)	0.35	12.1 (−1.5, 27.6)	0.07
7. Gonadal and related tumors	−0.5 (−1.5, 0.4)	0.19	2.6 (−0.3, 5.5)	0.07
7.1 Testis	−0.8 (−2.1, 0.5)	0.17	5.2 (−1.1, 11.9)	0.08
7.2 Ovary	1.3 (−0.3, 3.0)	0.09	−0.3 (−7.4, 7.3)	0.91
7.3 Germ cell and trophoblastic (GCT)-CNS	0.5 (−5.1, 6.5)	0.81	5.6 (−13.7, 29.2)	0.49
7.4 GCT (excluding CNS, testis, ovary)	−4.6 (−10.1, 1.2)	0.09	−1.1 (−8.8, 7.2)	0.72
8. Melanoma	−5.1 (−8.5, −1.6)	0.02	−7.4 (−14.2, −0.2)	0.047
9. Carcinomas	−0.7 (−3.0, 1.6)	0.43	−0.5 (−1.6, 0.7)	0.32
9.1 Thyroid carcinoma	−2.5 (−6.3, 1.5)	0.15	2.0 (−4.9, 9.5)	0.47
9.2 Other carcinoma of head and neck	−2.3 (−5.2, 0.7)	0.10	1.2 (−5.2, 7.9)	0.64
9.2.1 Nasopharyngeal carcinoma	−0.2 (−5.5, 5.5)	0.93	−0.3 (−17.2, 20.2)	0.97
9.2.2 Lip, oral cavity and pharynx	−3.0 (−6.0, 0.1)	0.06	1.6 (−8.5, 12.8)	0.69
9.2.3 Salivary gland	−1.9 (−5.0, 1.4)	0.18	5.6 (−10.2, 24.2)	0.40
9.2.4 Other carcinoma of head and neck	−2.7 (−11.6, 7.1)	0.47	−1.2 (−13.0, 12.3)	0.81
9.3 Carcinoma of gastrointestinal tract	1.1 (0.1, 2.0)	0.04	0.3 (−0.3, 0.9)	0.25
9.3.1 Carcinoma of esophagus	3.0 (−5.9, 12.9)	0.41	6.7 (−0.2, 14.1)	0.05
9.3.2 Carcinoma of stomach	−2.0 (−4.4, 0.6)	0.10	−0.8 (−4.9, 3.4)	0.61
9.3.3 Carcinoma of small intestine	2.3 (−4.3, 9.3)	0.40	−3.6 (−14.1, 8.1)	0.42
9.3.4 Carcinoma of colon	0.9 (−0.7, 2.5)	0.20	1.5 (−3.3, 6.5)	0.44
9.3.5 Carcinoma of rectum	2.7 (0.9, 4.6)	0.01	0.7 (−2.6, 4.2)	0.60
9.3.6 Carcinoma of anus	3.0 (−6.2, 13.0)	0.43	3.3 (−10.5, 19.3)	0.56
9.3.7 Carcinoma of liver and IBD	−1.4 (−6.6, 4.2)	0.52	−4.1 (−7.0, −1.2)	0.02
9.3.8 Carcinoma of gallbladder and other extrahepatic biliary	1.1 (−2.3, 4.6)	0.42	−0.9 (−15.9, 16.8)	0.88
9.3.9 Carcinoma of pancreas	2.2 (−4.2, 9.1)	0.40	2.6 (−4.5, 10.2)	0.38
9.4 Carcinoma of lung, bronchus, and trachea	−4.5 (−8.5, −0.3)	0.04	−2.8 (−7.6, 2.2)	0.19
9.5 Carcinoma of skin	−5.1 (−13.4, 4.1)	0.19	19.3 (−15.2, 68.0)	0.22
9.6 Carcinoma of breast	0.7 (−1.2, 2.6)	0.36	−1.8 (−5.0, 1.5)	0.20
9.7 Carcinoma of genital sites (non ovary or testis)	−1.5 (−4.7, 1.8)	0.26	−0.002 (−5.4, 5.7)	0.99
9.7.1 Carcinoma of uterine cervix	−3.2 (−6.5, 0.3)	0.06	−2.3 (−7.5, 3.2)	0.30
9.8 Carcinoma of urinary tract	−1.3 (−5.2, 2.8)	0.43	−3.6 (−7.4, 0.5)	0.07
9.8.1 Carcinoma of kidney	−0.7 (−4.6, 3.4)	0.66	−6.0 (−10.0, −1.9)	0.02
9.8.2 Carcinoma of bladder	−3.4 (−9.5, 3.2)	0.22	4.2 (−6.8, 16.5)	0.36
9.9 Carcinoma of other and ill-defined sites, NOS	4.2 (−0.3, 8.9)	0.06	1.8 (−2.5, 6.4)	0.31
10. Miscellaneous specified neoplasms	6.5 (−6.4, 21.2)	0.25	2.5 (−3.0, 8.2)	0.28
11. Unspecified Malignant Neoplasms, except CNS	−4.3 (−7.1, −1.5)	0.01	3.5 (0.1, 7.1)	0.046
Any cancer type, Total	−1.2 (−3.1, 0.8)	0.18	−0.48 (−1.0, 0.1)	0.07

^1^ APC: annual percentage change.

## Data Availability

Data is available on the SEER website, and the analytic methods, and Appendix A will be made available pending email request to the corresponding author.

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
