# Peer review of "Trends in Cancer Incidence and Mortality in US Adolescents and Young Adults, 2016–2021"

_cancers, 2024, doi:10.3390/cancers16183153_

Round 1

Reviewer 1 Report

Comments and Suggestions for Authors

The manuscript by Zhang L. et al. is interesting and well-executed. The authors adequately describe the frequency of cancer associated with the AYA group, as well as the risk factors involved. Despite the significance of this study, I have a few suggestions for the authors to review and address.

The authors need to review and better organize Table 1. I suggest aligning it with the WHO's classification of tumors and organizing it accordingly. For example, in the section on sarcomas, it would be advisable to subdivide soft tissue tumors and bone tumors. In the case of leukemias, include them under hematologic diseases, and so on. Do not group skin cancer and melanoma together. If possible, include the most important cities where the most frequent cancers occur. For example, lip cancer was reported with the highest incidence in Texas, particularly among young women, with squamous cell carcinoma being the most frequently reported variant. Please specify in Table 1 whether the malignant lesions of the lip, oral cavity, and pharynx were sarcomas or carcinomas, and replace "other carcinoma of the head and neck" with more specific classifications.

In the introduction, line 52, "Hodgkin's lymphoma" is repeated; please review this for accuracy.

Would it be possible to classify lymphomas more specifically? For example, you might note that diffuse large B-cell lymphomas were more common compared to other B-cell lymphomas, and that T-cell lymphomas were more frequent in women. Additionally, please provide some perspectives on the implications of this study.

Author Response

Response to the Comments and Suggestions for Authors

The manuscript by Zhang L. et al. is interesting and well-executed. The authors adequately describe the frequency of cancer associated with the AYA group, as well as the risk factors involved. Despite the significance of this study, I have a few suggestions for the authors to review and address.

Comment 1: The authors need to review and better organize Table 1. I suggest aligning it with the WHO's classification of tumors and organizing it accordingly. For example, in the section on sarcomas, it would be advisable to subdivide soft tissue tumors and bone tumors. In the case of leukemias, include them under hematologic diseases, and so on. Do not group skin cancer and melanoma together. If possible, include the most important cities where the most frequent cancers occur. For example, lip cancer was reported with the highest incidence in Texas, particularly among young women, with squamous cell carcinoma being the most frequently reported variant. Please specify in Table 1 whether the malignant lesions of the lip, oral cavity, and pharynx were sarcomas or carcinomas, and replace "other carcinoma of the head and neck" with more specific classifications.

Response 1: We thank reviewer’s comments and have revised table 1 accordingly in the revised manuscript, with the consideration for the fact that AYA cancer has a specific classification system which has been laid out in SEER. We did subdivide soft tissue tumors and bone tumors and made other revisions within the framework of SEER.

Comment 2: In the introduction, line 52, "Hodgkin's lymphoma" is repeated; please review this for accuracy.

Response 2: We thank reviewer’s comments and made modifications in the revised manuscript (Lines 52-55).

The incidence and types of cancer in AYA are age-dependent, with Hodgkin’s lymphoma, and central nervous system tumors more common in adolescents, whereas thyroid, testicular and non-Hodgkin’s’ lymphoma are relatively common in the 20–29-year range.

Comment 3: Would it be possible to classify lymphomas more specifically? For example, you might note that diffuse large B-cell lymphomas were more common compared to other B-cell lymphomas, and that T-cell lymphomas were more frequent in women. Additionally, please provide some perspectives on the implications of this study.

Response 3: We thank reviewer’s comments. Given that there are many kinds of lymphomas and that the low rates by multiple strata in trend analysis may be unstable, we further classify the lymphoma into Hodgkin lymphoma and non-Hodgkin lymphoma (NHL).  This is commonly used to monitor the cancer trends for AYA population.  Also, we detailed the gender differences in NHL and HL in the results section and compared our trends with the previously reported ones in the discussion section.

Reviewer 2 Report

Comments and Suggestions for Authors

From a biostats and clinical epidemiology point of view, here are some comments for the Authors:

- line 15 AYA, undefined

- line 108 age-adjusted incidence and mortality rates, IMHO it would be useful to add the MIR (mortality-to-incidence) ratio, as a proxy for AYA cancers survival

- line 137 Cancer was more common in women than in men, this info does not match with the usual clinical oncology metrics (normally, it`s true just the opposite), what do you believe!?

- line 138 The proportion of cancer deaths were similar in men and women, the same than above, moreover you should cite the number of death cases rather than the proportion (which is not a common epidemiological metric)

- line 141 Most cases occurred in metropolitan areas, these data have not been documented

- line 142 Age-adjusted AYA cancer incidence rates and mortality rates, it would be very useful to make a comparison with previous years SEER estimates, if available

- line 144 and following, per 100,000 may be omitted since it has been already defined

- line 145 most common diagnosed cancers, once again this is not a correct metrics, better to say new cases/incident cases

- table 1, add the MIR (mortality-to-incidence) ratio

- lines 162-187 p<0.05, please report the exact p-value with 3-sign digits, all around the manuscript

- line 163 most common, see comment at line 137

- lines 169-180 was significantly higher, how have you estimated significance!? please, provide details and stat tests

- line 186 nonmetropolitan areas, undefined, what do you mean?

- line 193, discrepancies is not a common epidemiological metric

- table 3, age-adjusted incidence and mortality trends from 2016-2021, lackness of details! Speaking about APC, maybe are you dealing with the ARC 'average Annual Rate of Change'? well, it`s undefined, moreover it represents the changes FROM 2016 TO 2021

- line 205, how have you estimated significance!? please, provide details and stat tests

- line 232 overall cancer mortality rates decreased for females, significantly?

- line 240 significant decline of incidence in 2020-2021, as compared to 2016-2019, how have you obtained this data? by which test?

- line 245 geographical disparity, is not a common epidemiological metric

- line 266 notable regional difference, what do you mean for regional?

- line 277 contrasted the previous trends, detail this statement!

- line 301 The main effect of Covid-19 in the general population is a decrease in major cancers, maybe you want to say that cancer incidence has been undereported due to the pandemy? please, clarify it

- line 335 RUCC, undefined

Author Response

Open Review

Responses to Comments and Suggestions for Authors

From a biostats and clinical epidemiology point of view, here are some comments for the Authors:

Comment 1: - line 15 AYA, undefined

Response 1: We thank reviewer’s comments and made the revision in the revised manuscript (Line 15) highlighted with track change and highlights:

Comment 2: - line 108 age-adjusted incidence and mortality rates, IMHO it would be useful to add the MIR (mortality-to-incidence) ratio, as a proxy for AYA cancers survival

Response 2: We thank reviewer’s comments and made the revision as follows in the revised manuscript (Lines 140-142) highlighted with track change and highlights:

Mortality-to-incidence ratio (MIR), a proxy for AYA cancer survival, was calculated by dividing the age-adjusted mortality rate by the age-adjusted incidence rate, for the selected cancers.

Comment 3: - line 137 Cancer was more common in women than in men, this info does not match with the usual clinical oncology metrics (normally, it`s true just the opposite), what do you believe!?

Response 3: We thank reviewer’s comments. While this is true in the adult population, breast cancer is the most common cancer in the AYA population. We made the revision as follows in the revised manuscript (Lines 145-147) highlighted with track change and highlights:

This study included 229,975 AYAs (aged 15-39 y) diagnosed with invasive cancer (women: 141,091 and men: 88,884) and 29,400 AYAs died from invasive cancer (men: 14,533 and women: 14,867) from 2016 to 2021 (Table 1 and Table S1).

Comment 4: - line 138 The proportion of cancer deaths were similar in men and women, the same than above, moreover you should cite the number of death cases rather than the proportion (which is not a common epidemiological metric)

Response 4: We thank reviewer’s comments and made the revision as follows in the revised manuscript (Lines 145-147) with track change and highlights:

This study included 229,975 AYAs (aged 15-39 y) diagnosed with invasive cancer (women: 141,091 and men: 88,884) and 29,400 AYAs died from invasive cancer (men: 14,533 and women: 14,867) from 2016 to 2021 (Table 1 and Table S1).

Comment 5: - line 141 Most cases occurred in metropolitan areas, these data have not been documented

Response 5: We thank reviewer’s comments and made the revision as follows in the revised manuscript (Lines 150-151) with track change and highlights:

Most cases occurred in metropolitan areas (metropolitan: 91.3% and nonmetropolitan: 8.6%).

Comment 6: - line 142 Age-adjusted AYA cancer incidence rates and mortality rates, it would be very useful to make a comparison with previous years SEER estimates, if available

Response 6: We thank reviewer’s comments. However, this is the result section where we reported the results generated from SEER.  We made the comparisons in the discussion section.

Comment 7: - line 144 and following, per 100,000 may be omitted since it has been already defined

Response 7: We thank reviewer’s comments and made the revision accordingly in the revised manuscript.

Comment 8: - line 145 most common diagnosed cancers, once again this is not a correct metrics, better to say new cases/incident cases

Response 8: We thank reviewer’s comments and made the revision as follows in the revised manuscript (Lines 154-155) with track change:

Thyroid and gastrointestinal tract carcinomas were the most common incident cases (15.3%) and cancer death (19.2%).

Comment 9: - table 1, add the MIR (mortality-to-incidence) ratio

Response 9: We thank reviewer’s comments and made the revision in the revised Table 1 with track change.

Comment 10: - lines 162-187 p<0.05, please report the exact p-value with 3-sign digits, all around the manuscript

Response 10: We thank reviewer’s comments and made the revisions in the revised manuscript with track change.

Comment 11: - line 163 most common, see comment at line 137

Response 11: We thank reviewer’s comments and made the revision as follows in the revised manuscript (Line 170) with track change:

For males, the common new cases were carcinomas (32.5%).

Comment 12: - lines 169-180 was significantly higher, how have you estimated significance!? please, provide details and stat tests

Response 12: We thank reviewer’s comments and made the revision as follows in the revised manuscript (Lines 112-116) with track change:

Comparison of the rates by race, sex, and metropolitan/nonmetropolitan designation were calculated using SEER*Stat 8.4.3.  The significance was estimated using the exact mid-P double sided P-value.

Comment 13: - line 186 nonmetropolitan areas, undefined, what do you mean?

Response 13: We thank reviewer’s comments, and we defined nonmetropolitan (Lines 100-102) as follows in the revised manuscript with track change:

while nonmetropolitan counties (RUCC code 4-9) by size of the largest city or town and proximity to metro and micropolitan areas (U.S. metro/nonmetropolitan population: 85.9% vs. 14.1%).

Comment 14: - line 193, discrepancies is not a common epidemiological metric

Response 14: We thank reviewer’s comments and made the revision as follows in the revised manuscript (Line 197) with track change:

Categorized by age group, the findings of the sociodemographic distribution of cancer incidence and mortality rates were mostly consistent, showing higher incidence rates in females vs males, NHW vs other racial groups (higher mortality was with NHB), nonmetropolitan vs metropolitan, with differing patterns in certain age groups (Table 2).

Comment 15: - table 3, age-adjusted incidence and mortality trends from 2016-2021, lackness of details! Speaking about APC, maybe are you dealing with the ARC 'average Annual Rate of Change'? well, it`s undefined, moreover it represents the changes FROM 2016 TO 2021

Response 15: We thank reviewer’s comments and we have described the details of the APC in the methods section (Lines 128-135).

Percent changes from 2016 to 2021 in age-adjusted incidence rates and in mortality rates were analyzed by fitting linear regression models with each calendar year at diagnosis or year of death (2016-2021) as an independent variable. Percent changes (significant increasing/decreasing or stable) in age-adjusted incidence and mortality rates from 2016-2021were assessed using APC calculated based on weighted least square methods and corresponding confidence intervals (CIs) estimated by SEER*Stat 8.4.3. The APC values allow for comparisons between incidence and mortality rate changes in rare and common cancers.

APC is defined in SEER to estimate the time trend of age-adjusted incidence/mortality rates from 2016-2021. https://seer.cancer.gov/help/seerstat/equations-and-algorithms/trend-algorithms

Comment 16: - line 205, how have you estimated significance!? please, provide details and stat tests

Response 16: We thank reviewer’s comments.  We estimated the significance of APC, and we reported p-value (Line 211-213). p=0.18 indicates that there was no significant percent change in overall age-adjusted incidence rates from 2016-2021.

There was no significant percent change in the overall age-adjusted cancer incidence rates (APC=-1.2, 95% CI: -3.1, 0.8, p = 0.18)

Comment 17: - line 232 overall cancer mortality rates decreased for females, significantly?

Response 17: We thank reviewer’s comments and made the revision (Line 239-240) as follows in the revised manuscript with track change.

The overall cancer mortality rates significantly decreased for females (APC = -0.8, p = 0.02). In women, a significant reduction was found for leukemia (-2.6, p = 0.03).

Comment 18: - line 240 significant decline of incidence in 2020-2021, as compared to 2016-2019, how have you obtained this data? by which test?

Response 18: We thank reviewer’s comments, and revised in the methods manuscript (Lines 114-116) as follows:

Using SEER*Stat 8.4.3, the age-adjusted incidence rates comparison between two periods: 2020-2021 and 2016-2019 (reference) was done in cancer subtypes. The significance was estimated using the exact mid-p double sided p-value.

Comment 19: - line 245 geographical disparity, is not a common epidemiological metric

Response 19: We thank reviewer’s comments and made the revision as follows (Line 252) in the revised manuscript with track change:

Figure S1 depicts the metropolitan/nonmetropolitan differences in cancer incidence trends.

Comment 20: - line 266 notable regional difference, what do you mean for regional?

Response 20: We thank reviewer’s comments and made the revision as follows (Lines 274-275) in the revised manuscript with track change:

The mortality trends from nerve sheath tumors differed between metropolitan and nonmetropolitan areas, with an increase in metropolitan areas (APC = 16.7, p = 0.03), and a fall in nonmetropolitan areas (APC = -23.2, p = 0.009).

Comment 22: - line 277 contrasted the previous trends, detail this statement!

Response 22: We thank reviewer’s comments and made the revision as follows in the revised manuscript (Lines 283-286) with track change:

The falling incidence trend of melanoma and male NHL reported in our study contradicted the previous finding where the melanoma incidence rates increased drastically, and trend of male NHL was stable from 1973-2015.

Comment 23: - line 301 The main effect of Covid-19 in the general population is a decrease in major cancers, maybe you want to say that cancer incidence has been undereported due to the pandemy? please, clarify it

Response 23: We thank reviewer’s comments and made the revision as follows (Line 311) in the revised manuscript with track change:

This indicates that cancer incidence may be underreported due to pandemics.

Comment 24: - line 335 RUCC, undefined

Response 24: We thank reviewer’s comments and the definition can be found in the material and method section (Line 98) in the revised manuscript.

Round 2

Reviewer 2 Report

Comments and Suggestions for Authors

The Authors were able to solve all previous concerns